# Factors Affecting Gut Microbiota of Puppies from Birth to Weaning

**DOI:** 10.3390/ani13040578

**Published:** 2023-02-06

**Authors:** Fatemeh Balouei, Bruno Stefanon, Sandy Sgorlon, Misa Sandri

**Affiliations:** Department of Agri-Food, Environmental and Animal Sciences, University of Udine, 33100 Udine, Italy

**Keywords:** pregnancy, microbiota, nutrition, metabolites, puppies

## Abstract

**Simple Summary:**

The literature in this review paper provided a recent perspective on the gut microbiome of newborn puppies. It was generally accepted that microbial colonization of newborns begins at birth, but some studies have reported evidence that suggests a healthy uterus is not always sterile and that the fetus at term already harbors bacteria. The microbiota development can occur during fetal life and can later be shaped by the type of parturition, colostrum, milk and lactation, environment, and food interactions. The studies suggested that the type of delivery, the maternal microbiota and their metabolites, and the diet of the mother influence the microbial community of a litters’ gut microbiota, which could be an important factor for the survival of newborns. The importance of the microbial composition of the mother for the development and growth of puppies deserves further study.

**Abstract:**

The review described the most important factors affecting the development of the intestinal microbiota in puppies from birth to weaning. The health and well-being of the microbiome in puppies is influenced by the type of parturition, the maternal microbiota, and the diet of the mother, directly or indirectly. The isolation of bacteria in dogs from the placenta, fetal fluids, and fetuses suggests that colonization could occur before birth, although this is still a matter of debate. Accordingly, newborn puppies could harbor bacteria that could be of maternal origin and that could influence microbial colonization later in life. However, the long-term impacts on health and the clinical significance of this transfer is not yet clear and needs to be investigated. The same maternal bacteria were found in puppies that were born vaginally and in those delivered via cesarean section. Potentially, the relationship between the type of parturition and the colonization of the microbiome will influence the occurrence of diseases, since it can modulate the gut microbiome during early life. In addition, puppies’ gut microbiota becomes progressively more similar to adult dogs at weaning, as a consequence of the transition from milk to solid food that works together with behavioral factors. A number of researches have investigated the effects of diet on the gut microbiota of dogs, revealing that dietary interference may affect the microbial composition and activity through the production of short-chain fatty acids and vitamins. These compounds play a fundamental role during the development of the fetus and the initial growth of the puppy. The composition of the diet fed during pregnancy to the bitches is also an important factor to consider for the health of newborns. As far as it is known, the effects of the type of parturition, the maternal microbiota, and the diet on the microbial colonization and the long-term health of the dogs deserve further studies. Definitely, longitudinal studies with a larger number of dogs will be required to assess a causal link between microbiome composition in puppies and diseases in adult dogs.

## 1. Introduction

The extensive access to next-generation sequencing technologies and their progressively cheaper cost have stimulated scientists to deepen the composition of microbial populations in different parts of the human and animal organisms. After a decade of intensive research (2007–2016), the human microbiome project produced evidence concerning the main sites of microbial colonization and demonstrated their relevance for the health and diseases of human beings, including the definition and characterization of enterotypes. 

Studies on fecal microbiome in domestic monogastric animals have highlighted the prevalence of a handful of phyla, which differ among species more for the abundances than for the taxonomic phylogenetic tree. Firmicutes, Bacteroidetes, and Proteobacteria are common in pigs, dog, cats, and humans [1], but carnivores show a higher presence of Fusobacteria and Actinobacteria [2,3]. However, these phyla have different abundances between humans and dogs [4], and also within dog populations, a high variability has been described [5,6], which also depends on the diet [7]. Part of the observed differences can be related to the sequencing technique, from DNA barcode or whole genome shotgun approaches, to upward and downward steps for feces collection, preservation, and methods for DNA extraction. The length of the reads, the selected primer for the hypervariable regions for the DNA barcode, the bioinformatic pipeline, and the reference annotation databases are also variables that can contribute to the reported differences [8]. Nevertheless, a comparison of the studies in gut microbiome can be considered a starting point to depict which are the more relevant factors affecting the gut microbial population.

The abundance and compositions of gut microbiota largely influence the physiology and health of the host, either due to the community composition or to the production of intermediate and end products of microbial metabolism. The invasion of pathogens directly affects the inflammatory and immune response of the host, but also, leads to a prevalence of some micro-organisms in the community. The reduction of the biodiversity can cause dysbiosis and inflammatory bowel disease (IBD). Related to the microbiome composition and under the pressure of environmental factors, nutrition, lifestyle, sex, and age, the concentrations of several compounds in the gut can vary. Some of these molecules have endocrine and neurotransmitter activities, and others have a nutritional and regulatory role in the physiology of the host. What it is clear now, is that the early microbial colonization of the gut in humans is important during pregnancy [9] since it has a short-term influence on the health and growth of the newborn and a long-term action later in life. Reported diseases related to microbiomes are, among others, food allergies, obesity, Type 2 diabetes, endocarditis, Alzheimer’s, and cancer [10,11,12,13,14].

The relationship between microbiomes and diseases can depend on the stimulation of the immune system of the host by antigens from the intestinal lumen [15,16], which can be one cause of the inflammation of the intestinal mucosa [17]. Following intestinal dysbiosis, genes that encode specific proteins can be down- or up-regulated under the influence of the altered environment [18]. The development of acute and chronic gastrointestinal diseases in dogs is highly related to gut dysbiosis [19,20,21]. Among the other diseases, canine atopic dermatitis is a disorder that is characterized by pruritus and secondary lesions of the skin, and it is due to environmental allergens [22] that stimulate the IgE antibodies-mediated response. Intestinal disbyosis has been reported to be associated with canine atopic dermatitis [23]. Additionally, neurodegenerative [24] diseases recognize the involvement of gut microbiomes in dogs.

The purpose of this review was to report the route of microbial translocation, if any, from the mother to the newborn during pregnancy, and the environmental factors that regulate the establishment of gut microbiomes in the puppies from birth to weaning. Additionally, the relevance of the nutrition of the mother during pregnancy and her lactation in shaping the microbial populations of the mother and the consequences that the compounds derived from the microbial metabolisms of the puppies were discussed. 

## 2. The Sterile Womb Paradigm: Still a Matter of Debate

The embryo and the fetus were considered to develop and growth in a sterile environment until the last century, meaning that the microbial colonization of the newborn happens only when it crosses from the maternal birth canal (Figure 1). Colostrum, interaction with the mother, milk and lactation, the environment, and foods contribute later to the shaping of the microbiome of the host. In healthy term pregnancies, sequencing of 16S rRNA indicates that amniotic fluid is sterile [25,26]; however, He et al. [27] found the presence of bacterial DNA in this fluid. Other DNA-based studies found that the uterus, placenta, amniotic fluid, meconium, and umbilical blood contain bacterial communities [28,29,30,31,32,33,34,35,36], suggesting a revision of the sterile womb paradigm for healthy pregnancies. The microbial contamination of the reproductive trait (follicular fluid, oviduct, and endometrium), including amniotic liquid and placenta during pregnancy, is still a matter of debate, and was discussed in recent reviews on animals [37,38] and human [39]. Nevertheless, DNA contaminations of reagents from laboratories are known artifacts affecting the sequence-based data of microbiomes [40].

The interest for the origin of the microbiome, if any, in the embryo and fetus arises from the importance that the initial colonization has on the establishment of the newborn gut microbial community and on the consequences that this priming population have on eubyosis and dysbiosis later in life and on the consequent development of diseases [41,42]. The way that these initial microbial translocated community affects the colonization of bacteria after birth, and through the early life, there is a point that requires investigation. Competitive exclusion is a term that refers to a process that prevents the colonization of other bacteria and is an interesting aspect to consider if the priming of microbiota, before the birth influences the inclusion or exclusion of specific taxa. A phylogenetic mathematical model was recently proposed to predict when and why new taxa are recruited into microbiota of infants [43]. The model indicated that species’ recruitment prevailed if a close relative was previously detected and that the competitive exclusion of phylogenetically close species was not relevant. In humans, it is known that the development of disease during life is linked to the gastrointestinal microbiota composition and colonization in early life [28,44]. 

The question is, where do these bacteria come from? The presence of bacteria in the blood of healthy animals has been reported in humans [45,46], and, although they are in a latent phase, they can shed in tissues and organs [47,48]. The placenta represents a barrier to bacteria and protects the fetus from microbial pathogens. However, bacteria have been documented in the basal plate of the human placenta [49] and cultivable bacteria were found in the amniotic fluid of mare [50]. The structure of the placenta differ among mammals and this can influence the transmission of bacteria from the blood to the fetus. However, in animals with a different structure of placenta, as cows [51,52,53], giant pandas [54], and dogs [55], the existence of an endometrial microbiome has been reported. The transplacental translocation of bacteria has been observed in mice [56] and also in sheep [57], and it can follow a dendritic cell route that is independent form the M epithelial cells that are located in the Peyer’s patches [58]. 

It is likely that the colonization of newborns starts in the uterus and, accordingly, bacteria causes the contamination of meconium [59,60]. Some of the very common bacteria that were isolated from the placenta and meconium of the puppies were *Staphylococcus* spp., *Streptococcus* spp., and *Neisseria zoodegmatis* [61]. Interestingly, these species were also present in the oral cavity and the vagina of the bitches. The latter authors were able to isolate the bacteria from 57% of the placental samples that were collected from the proximity of the chorioallantoic sac. However, samples were not collected from the basal plate, and this was likely the reason why no bacteria were found. Both [59,60] used a MALDI-TOF equipment to identify bacteria after their cultivation in an enriched medium, a technique that was developed for clinical purposes. In this case, the results refer to the presence of a bacterium, rather than its amount in the sample in relation to the whole microbial population, and the calculation of the relative abundance was not feasible. Moreover, this means that only culturable bacteria are detected, whilst other bacteria that can be often present in the gut for normal puppy development are potentially missed, since they are not cultivable. Conversely, genomic techniques do not consider if the bacteria are viable, but from a genetic perspective of microbiota composition, the metagenomics and whole genome sequencing analysis are able to describe the taxonomic structure in a quantitative way. 

Several studies have shown that the delivery type may influence the composition of the gut microbiota of a newborn. The gut microbiota of babies that are vaginally delivered (VD) includes bacteria harboring in the vagina and in the gut of the mother [61,62]. In contrast, other studies [63,64] have evidenced that the gut microbiota of babies born by caesarean section (CS) resembles the microbiota found in the skin and in the mouth of the mother. However, in these studies the samples of the infants’ gut microbiota were not collected immediately after parturition, but a few days after birth. Biascucci et al. [65] suggested that the mode of delivery of the newborn was important for gut development and that infants delivered by CS had an unusual gut microbiota. However, Grönlund et al. [66] observed that mothers’ and infants’ gut colonization is also shaped later by breast feeding. However, whether the differences of microbiomes due to the type of parturition have an effect on the colonization of the gut and on the health of dogs, both during the postnatal period and in later phases of life, it is still not well-established. Again, longitudinal studies of competitive exclusion will be of assistance to elucidate the association between the type of parturition and health during the stages of life.

In dogs, different bacteria were isolated from the uterus, amniotic fluid, and meconium after elective (ten dogs) or emergency CS (five dogs) [59]. Using culturable techniques and MALDI-TOF, *Acinetobacter* spp., *Bacillus* spp., and coagulase-negative *Staphylococci* were frequently found while *Glutamicibacter* spp., *Macrococcus* spp., *Pseudomonas* aeruginosa, *Micrococcus* spp., *Moraxella* spp., *Psychrobacter* spp., and *Stenotrophomonas* spp. were only occasionally identified. These data confirmed that uteruses and fetuses may not be sterile in healthy term canine pregnancies. The effect of parturition was also investigated by Zakošek Pipan et al. [60], who compared the microbiota of the placenta and meconium of puppies delivered by VD (26 puppies) and CS (elective 48 puppies; emergency 22 puppies). Swabs of the oral and vaginal mucosa were taken from the bitches in the second half of the pregnancy and just before parturition, and a culturable technique and MALDI-TOF analysis were performed. The same bacterial isolates were shared among puppies and their mothers in about 65% of the CS and in 100% of the VD. Moreover, in 40% of the parturitions, the bacterial species were the same in the oral microbiota of the mother and in the meconium of the newborn, while in 45% of the parturitions, the same bacteria were present in the vaginal mucosa and meconium. In this research [60], the puppies were stimulated to defecate just after the first intake of colostrum. This time span could have caused a bacterial colonization after birth, since this happens rapidly after delivery, as confirmed by Perez-Muñoz et al. [67]. The same critical factor of a time-lapse can be claimed in humans. He et al. [27] found a meconium microbiota within a couple of hours from birth after analyzing the first-pass meconium. 

The relationship between the type of parturition and the colonization of microbiomes in the postpartum period is another important aspect in relation to the modulation of the intestinal microbiome during the initial phase of life. In dogs, the colostrum is critical for the survival of puppies during the first weeks, since a layer of maternal endothelium separates the fetal chorion from the maternal blood (endotheliochorial placenta). The transplacental exchange of immunoglobulins mainly occurs during the last third of gestation in a specific placental region with a hemochorial organization [68]. For this reason, only about five to ten percent of immunoglobulins cross the placenta, and the puppies are nearly hypogammaglobulinemic at birth. In humans, colostrum is a source of bacteria [69], with a prevalence of *Streptococci*, *Lactobacilli,* and *Bifidobacteria*, which contribute to the initial colonization of the gut with consequences on the health and growth of the newborns.

In dogs, the microbiota of the colostrum from bitches that were delivered with an emergency CS was richer in *Staphylococcus* spp., *Lactobacillus*, *Kocuria,* and *Enterococcus* [70]. Instead, in elective CS and VD, *Staphylococcus pseudintermedius* was one of the most abundant bacteria, a species that was the major contributor of the oral and vaginal microbiota of dams [71]. This bacterium is a common inhabitant of the skin and is viewed as an opportunistic pathogen in dogs [72], and the isolation in the colostrum may therefore be due to contamination. Additionally, in humans, the presence of microbes in the colostrum can be due to the contamination of the oral cavity of the newborn and the surface of the skin of the mammary glands [73]. Another route is called the entero-mammary pathway, which is a translocation by the dendritic cells from the intestinal epithelial to the mammary glands via the lymphatic circulation [74]. 

Some question on the sterile womb paradigm are still to be resolved. In the case of dystocia and CS when the cervix was dilated, bacteria were isolated from the uterus after the extraction of all the puppies and fetal membrane [75], but *Micrococcus* spp. and *Macrococcus* spp. were isolated in a low-load and were not classified at the species level with biochemical tests alone. Moreover, these genera have been detected in the air of human and veterinary hospitals [76,77] and as contaminants of venous catheters [78]. Gobeli Brawand et al. [79] revealed that *Macrococcus* spp. is also isolated from the skin of healthy dogs and that it forms infection sites, and that *Staphylococcus equorum*, *S. hominis*, *S. saprophyticus* and *S. epidermidis* occur very commonly as commensals on the animal skin, rectum, and genitourinary tract [59].

The observed controversies might be the result of the small sample size, time of sampling, procedures adopted for the collection and storage of the samples, the techniques used for bacterial culturing and the DNA extraction methods, experimental contamination, and other factors. The isolation of bacteria from reproductive traits in dogs (the placenta, amniotic fluid, and fetus) can be considered as evidence that colonization begins before birth, and that the microbiota that originates from the mother can influence the development and health of the newborn and control colonization later in life. However, a published controversy involving outstanding researches in the field of microbiome studies [80] considers that more experiments are needed before definitive conclusions on the origin and extent of microbial contamination of reproductive traits can be drawn (Figure 2).

Although the transfer of microbiota from the mother to the gut of newborns during fetal life was reported, the real effect on the development of the microbial populations after birth and during weaning is not well-established yet.

## 3. Development of Microbiome in Puppies

The microbial colonization of the intestine after birth in humans [81,82] and dogs [83,84] is influenced by the mother, and the environment and can affect the health of the infants [85] and puppies [86]. However, the amniotic membrane in dogs is intact after the birth of a puppy, and in non-assisted parturition, the sac is broken by the teeth of the bitch. For this reason, the vaginal canal in dogs probably does not significantly contribute to the neonatal microbiota of the puppies that are more affected by lactation and the local environment. Differences in sterility during parturition and the hygiene conditions of living between humans and dogs are other important players in the process of microbial colonization. It is already accepted that breast milk is one of the most significant sources of microbes for newborn babies and infants after vaginal birth in humans [87], not only from the contact with the skin of the mother but also from the presence of microbes in the milk that can translocate from the intestine to the mammary ducts, which refers to the entero-mammary pathway [88].

There are fewer studies investigating the variations of bitch and puppy gut microbiomes from parturition to weaning and later in life. In the study by Del Carro et al. [84], vaginal, colostrum, and milk samples were collected from bitches after delivery and 48 h post-partum, and fecal samples were collected from dams and puppies at delivery and after day 2, 30, and 60 post-partum. Since the environment and the diet were the same, the authors concluded that the dams differed from each other concerning the microbiomes and this was shown in the fecal microbial compositions among the litters. The lower diversity within the litters than between the litters suggested that bitches prime the initial microbiota of puppies, but no data were collected during the growth and adult life of the newborns. Another conclusion from this study was that the variations among the litters decreased over time. These results referred to culturable bacteria, which represent a limited picture of the genetic biodiversity of the samples. Guard et al. [86], using 16S rRNA DNA sequencing, confirmed a progressive increase in microbial diversity and richness in the feces from 2 to 56 days of life. However, the fecal microbiota of puppies was significantly different from their mothers, and a clear association of the dam litter’s fecal microbiome was not reported. Another study based on 16S rRNA sequencing [89] confirmed that the fecal microbiome of 7-week-old German Shepherd puppies were more similar to that their own bitches than that of unrelated bitches, and that puppies of the same litter shared a less biodiverse microbiome in comparison to the unrelated puppies. There was a large animal to animal variation in the microbiome, and for the puppies, the variability also depended on the breed, age, diet, and environment. The dysbiosis index in puppies started to decrease after weaning, with values more similar to the reference value of adult dogs [90]. Specifically, the abundance of *Clostridium difficile* decreased and *Clostridium hiranonis* increased after this age, and this variation had a positive effect on the secondary bile acid concentration. Weaning is another milestone for microbiome compositions, due to the behavioral changes relating to the increase in exploratory activity [91] and often to the separation of puppies from their mother. The change in diet from liquid to solid and the variations of nutrients, including structural and non-structural carbohydrates, induces an expected shift of the gut microbiome. For dogs, a follow up from birth to adulthood, including weaning, confirmed that major changes of the microbiome occur around weaning [90], but continue until adult age. In another research, the relative abundance of Firmicutes and Actinobacteria was reduced from the age of 20 to 52 weeks old and Bacteroidetes increased [92], but the puppies were not sampled before weaning. Investigations in this direction are also required.

The clinical relevance of the microbiome composition at an early stage of life with the health at adult and geriatric ages of dogs is not well-investigated yet, and we are limited to the studies by Vilson et al. [89], Blake et al. [90], and Pereira et al. [92]. Longitudinal studies from birth to adulthood that aim to relate health with microbiome composition are also difficult to realize in humans, since environmental factors, individual genetic makeup, and exposure to pathogens are confounding factors. However, studies in humans have demonstrated that allergic asthma [93], dermatitis [94], metabolic diseases [95], and obesity [96,97] were related to the microbiome development. The pioneering study by Strachan [98] led to the ‘microflora hypothesis’ formulated later by Nover and Hugffnagle [99]. These authors suggested that the excess of hygiene, the use of antibiotics, and the limited contact with farm animals, due to urbanization, especially during childhood, limits the exposure to commensal microbiota and has the consequence of increased incidences of dysbiosis. As a result, oral and intestinal tolerances can be reduced with a shift toward a Th2 response, which contributes to an increase in allergy incidences [100]. Adaptive immunity develops in mammals after birth and the immune education continues throughout life. Oral and gut mucosa are continuously exposed to micro-organisms and food allergens and the immune system discriminates the self from the non-self to organize a defense and surveillance. To be tolerant to the non-self, T cells require instructions to eliminate self-sensitized cells or to convert them to tolerogenic Treg cells. The exposure to a more biodiverse commensal microbiome contributes to the generation of Treg cells [101]. In mice, the oral administration of *Clostridia* strains selected from human microbiota induced an increased number of Treg cells and the reduction of colitis and allergic diarrhea. Not only bacteria per se but also the product of fermentation, namely, short-chain fatty acids, contribute to oral tolerance and reduce food allergy [102]. In particular, a reduced butyrate concentration by microbiota fermentation in the gut enhances allergic sensitization during infancy [103]. Even though an interaction between the microbiota composition from birth to weaning and the development of disease later in life is not still clearly demonstrated, the initial colonization of the mouth and gut with micro-organisms has a paramount relevance for the immune development and response.

### 3.1. Effects of Maternal Diet on Puppies’ Microbiomes

Other aspects that deserve further investigation are the role of nutrition and the environment on the maternal microbiome, either during pregnancy or lactation. Nutrients supply and gut health during pregnancy are pivotal for the immune system of the bitch and for the transfer of passive immunity to the puppies with colostrum soon after birth. The role of diet in shaping intestinal microbiomes in dogs has been widely demonstrated [3,7,104], and nutrients and metabolites derived from the microbial fermentation of foods in the gut contribute to the normal development of a fetus and newborn. 

Several researches have investigated the effect of nutrition during pregnancy on the health of puppies, focusing on prebiotics and probiotics. Prebiotics are a class of nondigestible carbohydrates, including fructo-oligosaccharides, galacto-oligosaccharides, and manno-oligosaccharides, that are a substrate of beneficial micro-organisms. The fermentation of prebiotics by the intestinal microbiota regulates pH, reduces the competition of pathogens, and yields end products that are absorbed in the bloodstreams where they can exert a positive function. Indeed, probiotics are live micro-organisms that enhance the gut health of the host, restoring or improving the microbiomes. Gastroenteritis is a frequent condition among puppies, and the administration of prebiotics [105] and probiotics [106], *Lactobacillus rhamnosus* and *Lactobacillus plantarum* [107], can be used for these disorders. Additionally, the administration of prebiotics together with probiotics (symbiotic) to the bitches during pregnancy led to a reduction of gastroenteritis cases in puppies [108], mainly as a consequence of a colostrum with higher immune characteristics. In facts, the symbiotics showed a positive effect of gut immunity, due to the stimulation of IgA production [109]. However, the choice of probiotics for dogs in Europe, but not in the US, is limited by the EU Regulation 1831/2003, and *Enterococcus faecium* is the main bacterium admitted in pet food that possesses activity against enteropathogens, *Clostridium* spp., *Escherichia coli*, *Salmonella* spp., and *Shigella* spp. and [110].

During embryogenesis, the colonization of the gut microbiomes of the puppies can play a role in the in the formation of an enteric network, including vagal ganglia and glia differentiation, the innervation of the gut by the sympathetic nerve [111]. After birth, the microbiota also exerts an early postnatal control of the intestinal stem cells proliferation and differentiation, increasing the length of the villi, vascularization, the development of the local immune system and enzymes, and the completion of the neuronal functions and mucosal glia, which continues during weaning together with a decrease in mucosa permeability [111]. There are compelling pieces of evidence that suggest that gut microbiota plays a paramount role in the development of the enteric nervous system and circuitry in animals [112,113,114], and, consequently, the administration of probiotics can shape this process [115,116]. 

The way that the microbiota interacts with the host in shaping nervous systems and for the orchestration of local and systemic immune responses is also mediated by a battery of metabolites that are produced by the different microbes inhabiting the organism. The metabolites and the compounds derived from the fermentations of the mother microbiota can enter in the fetus [117]. Among those, the end products of carbohydrates and amino acids fermentations, notably short straight- and branched-chain fatty acids (SCFAs), lactic acids, and the metabolism of vitamins, specifically folate, Vitamin B12, and choline, will be considered.

### 3.2. Short-Chain Fatty Acids (SCFA)

The short-chain fatty acids are microbial metabolites mainly originating during the colonic and cecal anaerobic fermentation of structural and non-structural carbohydrates, and, to a lesser extent, amino acids. Diet composition and microbiota largely influences the yield of SCFAs, lactates, and their molar proportions. In dogs, the data obtained in our laboratory (Figure 3) show that the molar proportion of acetate, propionate, isobutyrate, and butyrate are on average 47.1%, 25.7%, 10.1%, and 8.8%, respectively, with a minor proportion of lactate, isovaleric, and valeric [2,3,7,118]. The molar proportion of the branched-chain fatty acid isobutyrate ranges from 2 to 15%, for isovalerate from 1 to 3%, and for lactic acid from 3 to 15%. Lactic acid and SCFAs in the feces can differ from the concentrations in the bowel in relation to the colonic absorption and to the local consumption in the gut [119]. 

In mice, it has been reported that the effect of diet on microbiota and their metabolites influences the incidence of asthma in the offspring [120], and it was found that acetate reduces airway allergic diseases due to the increase in the number of functions of T-regulatory. Enteric SCFAs enhance the proliferation of neural progenitor cells in humans and it is plausible that the transfer of these compounds from the mother to the embryos and to the fetus can have an important regulatory role during the development of newborns [121]. SCFAs have regulatory roles other than contributing to the nutrient supply of the host, due to the interaction with specific cell receptor, free fatty acids receptor (FFAR). 

The regulatory activity of SCFAs on cell functions depends on the presence of specific cellular receptors (G protein-coupled receptors, GPCRs), which regulate several cellular processes and modulates the environmental stimuli to adapt to the biological response of cells [122]. GPCR40 (alias FFAR1) is activated by medium-chain fatty acids and long-chain fatty acids and regulates insulin secretion in pancreatic islet [123,124]. GCPR120 (alias FFAR4) is activated by omega-3 fatty acids, as linoleic, eicosopentaenoic, and docosahexaenoic acids and is involved in anti-inflammatory effects [125,126]. From a host microbiome crosstalk perspective, GPCR43 (alias FFAR2) and GPCR41 (alias FFAR3) gained attention for their sensitivity to SCFAs [101,117,127]. For FFAR2, the rank of ligand potency is propionate = acetate > butyrate > valerate, and for FFAR3, propionate = butyrate > valerate > acetate [128,129]. 

Between the SCFAs, butyrate has attracted the attention of researchers for its regulatory role of host physiology, since it stimulates the mitochondrial uncoupling protein-1 (UCP-1) in brown adipose tissues, enhancing thermogenesis and fatty acid oxidation, and activates AMPK signaling in muscles [130]. Moreover, butyrate can interact with the nervous system [131], and regulates the sympathetic nervous system [132] and hippocampal neurogenesis [133] and activates vagal sensory neurons with food intake inhibition [134]. Interestingly, butyrate is also a potent inhibitor of deacetylases (HDACs), which have an epigenetic promoting acetylation of histone and stimulate gene expression in host cells. Acetylation of lysine in the histone tails, mainly at H3 and H4, promotes the opening of chromatin and activates gene transcriptions. Histone acetylation mediated by the administration of butyrate has been shown to influence neuronal plasticity, long-term potentiation of memory [135,136], and restore cognitive functions [137,138]. Moreover, butyrate was found to reverse a neurocognitive deficit in mice offspring [139]. 

Butyrate-producing microbes in humans belong to Firmicutes phylum, clostridium cluster I, IV, IX, XVI, and XIVa [140], a group of bacteria that does not refer to a taxonomic level, as a family of a genus. Firmicutes phylum is also predominant in the feces of dogs, [141] and, among the others, the *Clostridium*, *Eubacterium*, *Faecalibacterium,* and *Roseburia* genera are inhabitants of the gut in dogs [18]. Members of the Ruminococcaceae (*Faecalibacterium prausnitzii*), Lachnospiraceae (*Eubacterium rectale; Roseburia inulinivorans; Roseburia intestinalis*), and Erysipelotrichaceae (*Eubacterium biforme*) families are known to synthetize butyrate through the butyryl-CoA: acetate-CoA transferase pathway, although the butyrate kinase route is an alternative pathway that is activated in bacteria of other phyla [142].

### 3.3. ONE Carbon Metabolism 

Others compounds that derive from microbial metabolism are the vitamin B. A systematic genome assessment of vitamin B-producing bacteria [143] indicated a pattern of occurrence and distribution in human microbiomes.

Folate is involved in one carbon metabolism together with methionine and choline and after the reduction of tetrahydrofolate, the biologically active form is further converted to 5,10-methylenetetrahydrofolate and then to 5 methyltetrahydrofolate (Figure 4). Vitamin B12 is a cofactor of the conversion of 5,10-methylenetetrahydrofolate to 5 methyltetrahydrofolate, which functions as a methyl donor to homocysteine for the remethylation of methionine [144]. Alternatively, a methyl group can be donated from choline to homocysteine. The methionine and the folate cycles are related to the methylation of DNA, which is a main driver of the epigenetic regulator and the control of gene expression, and of RNA, proteins, and lipids. As a consequence, environmental and dietary factors that affect the ONE carbon metabolism can interfere with the programming of embryonic and fetal developments [145]. 

Like humans, dogs depend on folates and vitamins involved in the ONE carbon metabolism from dietary supplementation and their dietary regime, which also affects the synthesis from gut microbiomes, and the overall supply regulates the host’s physiology [146]. The analysis of folate and B12 concentrations in the serum of dogs has been proposed as biomarkers of the gut functions [118,147,148]. A decreased concentration of cobalamin and an increased concentration of folate are related to intestinal dysbiosis [149], a term referring to the imbalance of gut microbiotas that is associated with several diseases in dogs [19].

During pregnancy, the amounts and the type of fats and vitamins ingested can lead to a maternal proinflammatory microbiota [150], which has an impact on newborns. In Wistar rats, the excess of imbalance of folate and choline modified the mother’s microbiome, leading to obesogenic rat pups [151]. It has been reported that obesity during pregnancy negatively impacts the fetal development and well-being of newborns in humans [152], due to an alteration of the ONE carbon metabolism, caused by a shift in gut microbiota and a consequent decrease in vitamin B synthesis. 

In pregnant women, the gut microbiome compositions and functions are associated with preeclampsia, a disorder of pregnancy characterized by hypertension, protein in the urine, and the damage of organs. Among the biomarkers of preeclampsia, a significant downregulation of the KEGG pathways of folate metabolism, the metabolism of cofactors and other vitamins, pyridoxal-P biosynthesis, riboflavin metabolism, and cobalamin biosynthesis was found. Conversely, the microbiota resulted in an enrichment of the species belonging to the genera *Blautia*, *Pauljensenia*, *Ruminococcus,* and *Collinsella* [153].

## 4. Conclusions

According to studies and their data, we can consider that the colonization of the intestinal microbiota in puppies from birth to weaning is dynamically influenced by many factors. In this review, we proposed that the sterile womb paradigm requires a reconsideration and needs to be further studied, since it is an area of research that has not been investigated extensively, at least in dogs. The long-term effects on health of the colonization of gut microbiomes in relation to maternal transfer before birth, the type of parturition, the intake of colostrum and milk, and weaning is at the moment more speculative than proven. The diet of the mother during pregnancy can influence not only her own gut microbiota but provides the fetus with metabolites, such as SCFAs and vitamins, which interfere with the development of the fetus and the growth of the puppies. 

This is an area of study that deserves further investigation in dogs and the clinical significance of the colonization of gut microbiomes in puppies requires an evaluation by the means of longitudinal studies. Additionally, the studies reported here present limitations due to the reduced population size, which makes it difficult to obtain the required statistical power for the factors investigated and requires collaborative studies among the researchers.

## Figures and Tables

**Figure 1 animals-13-00578-f001:**
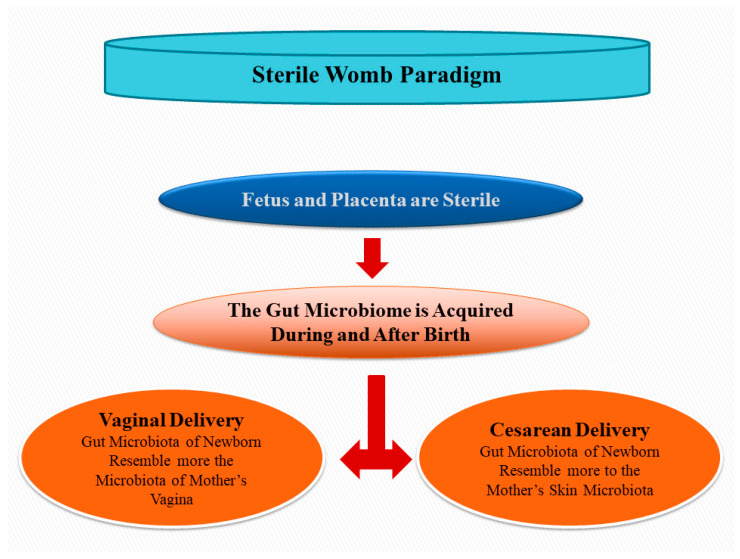
In the sterile womb paradigm, the placenta, amniotic fluid, and the gut of fetuses are not colonized by micro-organisms during pregnancy, and the intestinal microbiome of the newborn is acquired only during parturition and after the birth. Of note, the type of parturition affects early microbial colonization of the newborn.

**Figure 2 animals-13-00578-f002:**
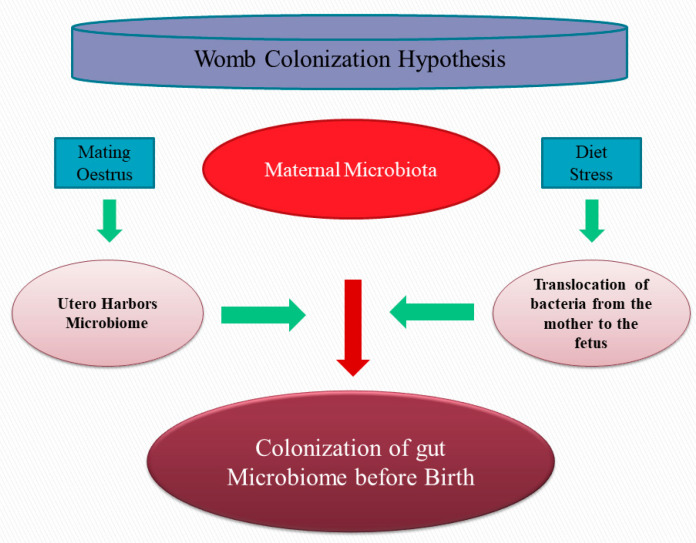
During pregnancy, the intestine of fetus is colonized by micro-organisms of the mother.

**Figure 3 animals-13-00578-f003:**
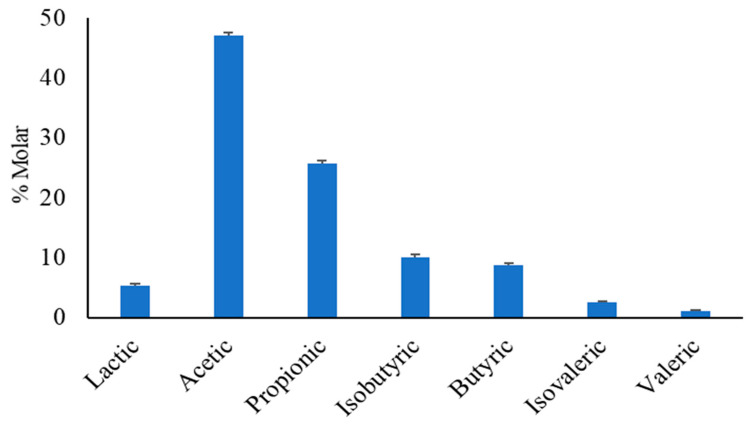
Mean and standard error of the molar proportion of short-chain fatty acids and lactic acid in the feces of dogs.

**Figure 4 animals-13-00578-f004:**
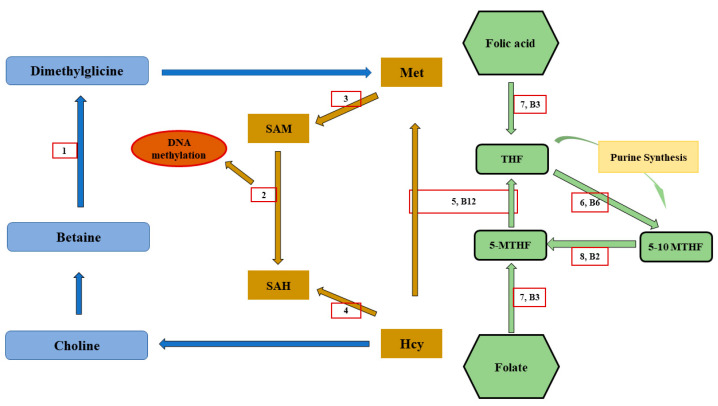
A simplified scheme of ONE carbon metabolism and its implication on DNA methylation. (1: betaine homocysteine methyltransferase (BHMT); 2: DNA methyltransferase (DNMT); 3: methionine adenosyltransferase (MAT); 4: S-adenosyl homocysteine hydrolase (SAHH); 5: methionine synthase (MS); 6: serine hydroxymethyltransferase (SHMT); 7: dihydrofolate reductase (DHFR); 8: methylenetetrahydrofolate reductase (MTHFR); SAH: S-adenosylhomocysteine; SAM, S-adenosylmethionine; THF: tetrahydrofolate; 5-MTHF: 5-methyltetrahydrofolate; and 5,10-MTHF: 5,10-methyltetrahydrofolate.) (adapted from [144]).

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
