# Peer review of "Factors Affecting Gut Microbiota of Puppies from Birth to Weaning"

_animals, 2023, doi:10.3390/ani13040578_

Round 1

Reviewer 1 Report

The review article ‘Factors Affecting Gut Microbiota of Puppies from Birth to Weaning’ - is very well written with nice figures and provides interesting information on the translocation of bacterial populations to puppies.

In this reviewer’s opinion - perhaps add some additional information (or another smaller section) that describes the impact -if  any - on factors such as a competitive exclusion, and the localized host innate and adaptive immunity (immunity within the body region) on the translocation and establishment of the microbiota within puppies

Author Response

We greatly thank the Reviewer for the comment and below the answers.

 The review article ‘Factors Affecting Gut Microbiota of Puppies from Birth to Weaning’ - is very well written with nice figures and provides interesting information on the translocation of bacterial populations to puppies.

In this reviewer’s opinion - perhaps add some additional information (or another smaller section) that describes the impact -if  any - on factors such as a competitive exclusion, and the localized host innate and adaptive immunity (immunity within the body region) on the translocation and establishment of the microbiota within puppies

AU: We really thank the reviewer for the suggestion and, in the text, we considerations about the competitive exclusion concept are reported (Lines 116-124). Also, the tolerogenic role of microbiome colonization for oral and gut health was briefly described (Lines 308-335). These two aspects are really of great importance and probably deserve a specific review for dogs.

Reviewer 2 Report

1) The purpose of this review was to report the route of microbial translocation, if any, from the mother to the newborn during pregnancy, the environmental factors that regulate the establishment of gut microbiome on the puppies from birth to weaning, therefore very important a literature review on this topic since there are still controversies the subject. 

2)Very important aspects well highlighted in this review: the uterus is not totally sterile  since fetal maternal bacterial translocation occurs and many neonates may already be born in sepsis; colostrum, interaction with the mother, milk and lactation, environment and foods contribute later in the shaping the microbiome of the host; the type of delivery interferes with the obtaining of the microbiome during and after birth, and there are differences according to delivery; 

3)I think more aspects would be important regarding the critical phase of weaning in puppies, I missed. 

4)Page 3, line 113: word dormient = latent (in Italian, please replace with English latent)

5)Page 4 figure= improve image quality (figure of the authors?)

6)Page 4, line 150: word pf = of

7)Page 4 line 170 = first intake of colostrum. this time span (uppercase)

8)Page 5 = figure 2 improve image quality (figure of the authors?)

9)Page 5, line 181 = figure 2 refers to the colonization of the fetal intestine during pregnancy, according to the legend, however, the sentence that begins at line 178, refers to the type of delivery and the postpartum aspects. I believe that Figure 2 should be inserted in another part of the text as a suggestion, page 6, line 223.

10)Page 5, line 187 = animmunoglobulinemic replace by agammaglobulinemic or hypogammaglobulinemic

11)Page 9, line 388: replace leading to obesogenic puppies with obesogenic rat pups

12)Page 10, line 427= "requires collaborative studies among the researchers" excelent proposition and very good review, congratulations to the authors

Author Response

Thank you to the reviewer for the valuable comments and suggestion, that we have replied below.

1) The purpose of this review was to report the route of microbial translocation, if any, from the mother to the newborn during pregnancy, the environmental factors that regulate the establishment of gut microbiome on the puppies from birth to weaning, therefore very important a literature review on this topic since there are still controversies the subject. 

2)Very important aspects well highlighted in this review: the uterus is not totally sterile since fetal maternal bacterial translocation occurs and many neonates may already be born in sepsis; colostrum, interaction with the mother, milk and lactation, environment and foods contribute later in the shaping the microbiome of the host; the type of delivery interferes with the obtaining of the microbiome during and after birth, and there are differences according to delivery; 

3)I think more aspects would be important regarding the critical phase of weaning in puppies, I missed. 

AU: Thank you for the suggestion, we have tried to expand this section (Lines 295-307), but there is a paucity of information on this delicate passage from liquid to solid diet.  

4)Page 3, line 113: word dormient = latent (in Italian, please replace with English latent)

AU: we modified as suggested.

5)Page 4 figure= improve image quality (figure of the authors?)

AU: the figure was drawn by the authors and the quality was improved.

6)Page 4, line 150: word pf = of

AU: we corrected the typo.

7)Page 4 line 170 = first intake of colostrum. this time span (uppercase)

AU: we corrected the typo.

8)Page 5 = figure 2 improve image quality (figure of the authors?)

AU: the figure was drawn by the authors and the quality was improved.

9)Page 5, line 181 = figure 2 refers to the colonization of the fetal intestine during pregnancy, according to the legend, however, the sentence that begins at line 178, refers to the type of delivery and the postpartum aspects. I believe that Figure 2 should be inserted in another part of the text as a suggestion, page 6, line 223.

AU: Thank you for the comment. The figure was moved as indicated.

10)Page 5, line 187 = animmunoglobulinemic replace by agammaglobulinemic or hypogammaglobulinemic

AU: we corrected as suggested

11)Page 9, line 388: replace leading to obesogenic puppies with obesogenic rat pups

AU: we corrected as suggested

12)Page 10, line 427= "requires collaborative studies among the researchers" excelent proposition and very good review, congratulations to the authors

AU: Thank you, hoping that a collaborative study will start soon.

Reviewer 3 Report

There is a disconnect between the statements in the abstract and the statements in the text (see comments below). The authors cite many review articles, but it is important to cite the original experimental studies

Lines 23-29: I think that there is not yet convincing evidence for this, so I suggest to provide more balanced language. Especially there is no evidence yet that this has a clinical significant impact on long-term health. The authors state a lot of potentials, but there are no mechanisms yet described if this is relevant at all (even in humans the studies are still ongoing whether that makes a longterm difference in health)

Line 29-31: there is also not much evidence in larger studies that puppies become more similar than mothers (they becoming more adult)

Line 34-35: there is not much evidence for probiotic that they truly would make a clinical important impact (except that some markers improve) – does that really change the long-term trajectory of animals?

Line 37-38 – this sentence is not clear what authors mean

Lines 47-52: this paragraph is welcomed, but at the same time it is crucial to add a major limitation of NGS – that sequencing has never been analytically validated for reproducibility etc.. Much depends on primer selection etc when comparing different studies. Almost all studies in veterinary medicine have examined a small number of animals and compared only few groups - so how large is the size effect of observed differences when compared to a large reference population and to diseased groups? Maybe the observed differences are very minor and clinically not relevant? This needs to be mentioned to bring to the reader a more balanced view of the data out there.

Line 80-81: the cited reference is a review article – can you provide here (but also everywhere else where you cite a review article) the original studies that support your statement?

Line 85: where is the evidence that microbiome is related to parvovirus? Yes they responded better to FMT, but how does that prove that microbiota and parvovisosis have a relationship? It could have been because puppies received antibiotics and that caused dysbiosis

Line 93: here you state it’s a matter of debate (which I agree), which is different than in the abstract

Line 104: you should provide a few sentences of this debate – that DNA could be also form reagent contamination etc.. This should help the reader to understand the debate

Line 110-111 – can you please expand – what exactly is the time period, what diseases are linked to changes? Is this in relation to presence of bacteria in the placenta etc… or do you mean general microbiota is linked to disease. It is important to differentiate in those paragraphs and state whether you refer to overall microbiota in early age (in intestinal lumen), or the findings of bacteria or DNA in the tissues which were thought to be sterile? So ideally you completely separate these two concepts into separate paragraphs for easier reading. So for example in lines 112-121 is there any clinical significance of this? Or are these just descriptions?

Line 155 – sentence is incomplete. Again, what is the clinical significance besides that gut microbiota is different – what is the size effects of these differences, maybe just statistical? At least mention that at this point it is not clear whether this is relevant for health

Line 156 and following - It would be useful in veterinary studies to always state the number of animals included and which techniques were used to give reader a better understanding

Line 181: what diseases has been proven – please be specific

·         Lines 255 -257; the paper Blake et al (PMID: 33047396) should be mentioned here, as it allows a good estimation in which week some of the important intestinal  bacteria (like C. hiranonis and Faecalibacterium) are within reference intervals of adult dogs and also bile acid metabolism becomes normal

Line 279 – should be mentioned that this is in Europe an not in USA

Line 290 – compiling?

Lines 320-345 – this paragraph is technically very complex and written in much detail in comparison to the remaining manuscript. It would be better to focus on SCFA and development of puppies, which at the moment is not the case

Line 383: low folate is more of a marker of malabsorption (high folate may indicate dysbiosis). Low cobalamin is also a marker of malabsorption. While often mentioned that low cobalamin can be due dysbiosis, there is no real clear published evidence – can the authors provide a original study rather than a review article citation?

Line 419: differences in microbiota alone may not be important – how should we study the clinical significance of this?

Author Response

We appreciated the comments of the Reviewer and the answers are reported in the following lines. 

There is a disconnect between the statements in the abstract and the statements in the text (see comments below). The authors cite many review articles, but it is important to cite the original experimental studies

AU: We thank the reviewer for the comments and the suggestions. We have chosen to quote review article to reduce the number of citation, but we agree with the suggestion of the reviewer and have now included the original articles. The abstract was revised accordingly.

Lines 23-29: I think that there is not yet convincing evidence for this, so I suggest to provide more balanced language. Especially there is no evidence yet that this has a clinical significant impact on long-term health. The authors state a lot of potentials, but there are no mechanisms yet described if this is relevant at all (even in humans the studies are still ongoing whether that makes a longterm difference in health)

AU: The text was modified

Line 29-31: there is also not much evidence in larger studies that puppies become more similar than mothers (they becoming more adult)

AU: The text was modified

Line 34-35: there is not much evidence for probiotic that they truly would make a clinical important impact (except that some markers improve) – does that really change the long-term trajectory of animals?

AU: The text was modified

Line 37-38 – this sentence is not clear what authors mean

AU: The text was modified

Lines 47-52: this paragraph is welcomed, but at the same time it is crucial to add a major limitation of NGS – that sequencing has never been analytically validated for reproducibility etc.. Much depends on primer selection etc when comparing different studies. Almost all studies in veterinary medicine have examined a small number of animals and compared only few groups - so how large is the size effect of observed differences when compared to a large reference population and to diseased groups? Maybe the observed differences are very minor and clinically not relevant? This needs to be mentioned to bring to the reader a more balanced view of the data out there.

AU: Thank you for the comments. We are aware that differences of sequencing techniques and upward and downward steps for faeces collection, preservation, DNA extraction, NGS platform and reads length, use of DNAbarcode or WGS, bioinformatic pipeline and reference annotation database, statistical analysis influence the results. The limitation of methodology applied and technology used is relevant, but it is our opinion that this will merit a specific review. In the conclusion we have comment that size effect and small population are limited in veterinary studies.

Line 80-81: the cited reference is a review article – can you provide here (but also everywhere else where you cite a review article) the original studies that support your statement?

AU: Original article was reported.

Line 85: where is the evidence that microbiome is related to parvovirus? Yes they responded better to FMT, but how does that prove that microbiota and parvovisosis have a relationship? It could have been because puppies received antibiotics and that caused dysbiosis

AU: We agree, reference to parvovirosis was deleted.

Line 93: here you state it’s a matter of debate (which I agree), which is different than in the abstract

AU: The abstract was reworded.

Line 104: you should provide a few sentences of this debate – that DNA could be also form reagent contamination etc.. This should help the reader to understand the debate

AU: We have added reference to DNA contamination. In this case we suggest to read the review to deepen the debate (Lines 109-111).

Line 110-111 – can you please expand – what exactly is the time period, what diseases are linked to changes? Is this in relation to presence of bacteria in the placenta etc… or do you mean general microbiota is linked to disease. It is important to differentiate in those paragraphs and state whether you refer to overall microbiota in early age (in intestinal lumen), or the findings of bacteria or DNA in the tissues which were thought to be sterile? So ideally you completely separate these two concepts into separate paragraphs for easier reading. So for example in lines 112-121 is there any clinical significance of this? Or are these just descriptions?

AU: The sentence at lines 110-111 refers to human and the time period are later described for puppies (lines 247-270). We have added sentences about the possible relationship of colonization in utero and in early life. The clinical relevance is reported in lines 112-140.

Line 155 – sentence is incomplete. Again, what is the clinical significance besides that gut microbiota is different – what is the size effects of these differences, maybe just statistical? At least mention that at this point it is not clear whether this is relevant for health

AU: As suggested the sentence was modified.

Line 156 and following - It would be useful in veterinary studies to always state the number of animals included and which techniques were used to give reader a better understanding

AU: As requested, the information was included in the text (Lines 180-191).

Line 181: what diseases has been proven – please be specific

AU: For dogs, longitudinal studies that relate the microbiota colonization and development in puppies with the health during life are lacking, and few information are available in humans and laboratory animals. The text was reworded (Lines 308-336)

Lines 255 -257; the paper Blake et al (PMID: 33047396) should be mentioned here, as it allows a good estimation in which week some of the important intestinal  bacteria (like C. hiranonis and Faecalibacterium) are within reference intervals of adult dogs and also bile acid metabolism becomes normal

AU: Thank you for the suggestion. The text was revised accordingly.

Line 279 – should be mentioned that this is in Europe an not in USA

AU: We reworded the sentence.

Line 290 – compiling?

AU: thank you, the word was corrected.

Lines 320-345 – this paragraph is technically very complex and written in much detail in comparison to the remaining manuscript. It would be better to focus on SCFA and development of puppies, which at the moment is not the case

AU: We have simplified the paragraph, to underline the way that maternal metabolites is related to development of newborns during pregnancy (Lines 401-434).

Line 383: low folate is more of a marker of malabsorption (high folate may indicate dysbiosis). Low cobalamin is also a marker of malabsorption. While often mentioned that low cobalamin can be due dysbiosis, there is no real clear published evidence – can the authors provide a original study rather than a review article citation?

AU: Thank you for the comment. The sentence was modified and the reference to research paper added (Lines 475-480).

Line 419: differences in microbiota alone may not be important – how should we study the clinical significance of this?

AU: The conclusions were reworded, to account for the comments of the reviewer.

Round 2

Reviewer 3 Report

Thank you for the revisions! The paper is now much more balanced and informative.

I have only one more comment, about the NGS methods. Your response (see below) should be appropriately reworded to fit the text and then added to that paragraph of the manuscript, as too many readers are not aware of these limitations. So the few sentences are sufficient but crucial to add

"We are aware that differences of sequencing techniques and upward and downward steps for faeces collection, preservation, DNA extraction, NGS platform and reads length, use of DNAbarcode or WGS, bioinformatic pipeline and reference annotation database, statistical analysis influence the results. The limitation of methodology applied and technology used is relevant, ....."

Author Response

Thank you for the revisions! The paper is now much more balanced and informative.

I have only one more comment, about the NGS methods. Your response (see below) should be appropriately reworded to fit the text and then added to that paragraph of the manuscript, as too many readers are not aware of these limitations. So the few sentences are sufficient but crucial to add

"We are aware that differences of sequencing techniques and upward and downward steps for faeces collection, preservation, DNA extraction, NGS platform and reads length, use of DNAbarcode or WGS, bioinformatic pipeline and reference annotation database, statistical analysis influence the results. The limitation of methodology applied and technology used is relevant, ....."

AU: Dear Reviewer, we provide the paragraph as request, hoping this will fit your comments.

Regards
